# Effectiveness of a behavioural intervention involving regular weighing and feedback by community midwives within routine antenatal care to prevent excessive gestational weight gain: POPS2 randomised controlled trial

Amanda Daley,[1] Kate Jolly,[2] Susan A Jebb,[3] Andrea Roalfe,[3] Lucy Mackilllop,[4] Amanda Lewis,[5] Sue Clifford,[2] Muhammad Usman,[2] Corah Ohadike,[6] Sara Kenyon,[2] Christine MacArthur,[2] Paul Aveyard[3]

For numbered affiliations see end of article.

**Correspondence to**
Professor Amanda Daley;
a.daley@lboro.ac.uk

## ABSTRACT

**Objectives** To assess the effectiveness of a brief behavioural intervention based on routine antenatal weighing to prevent excessive gestational weight gain (defined by US Institute of Medicine).

**Design** Randomised controlled trial.

**Setting** Antenatal clinic in England.

**Participants** Women between $10^{+0}$ and $14^{+6}$ weeks gestation, not requiring specialist obstetric care.

**Interventions** Participants were randomised to usual antenatal care or usual care (UC) plus the intervention. The intervention involved community midwives weighing women at antenatal appointments, setting maximum weight gain limits between appointments and providing brief feedback. Women were encouraged to monitor and record their own weight weekly to assess their progress against the maximum limits set by their midwife. The comparator was usual maternity care.

**Primary and secondary outcome measures** Excessive gestational weight gain, depression, anxiety and physical activity.

**Results** Six hundred and fifty-six women from four maternity centres were recruited: 329 women were randomised to the intervention group and 327 to UC. We found no evidence that the intervention decreased excessive gestational weight gain. At 38 weeks gestation, the proportions gaining excessive gestational weight were 27.6% (81/305) versus 28.9% (90/311) (adjusted OR 0.84, 95% CI: 0.53 to 1.33) in the intervention and UC group, respectively. There were no significant difference between the groups in anxiety or depression scores (anxiety: adjusted mean −0.58, 95% CI:−1.25 to −0.8; depression: adjusted mean −0.60, 95% CI:−1.24 to −0.05). There were no significant differences in physical activity scores between the groups.

**Conclusions** A behavioural intervention delivered by community midwives involving routine weighing throughout pregnancy, setting maximum weight gain targets and encouraging women to weigh themselves each week to check progress did not prevent excessive

## Strength and limitations of this study

- ► Most (80%) eligible women participated in the trial, meaning the results reflect the impact in the general population.
- ► A relatively large proportion of women were recruited from non-white ethnic groups and/or low socio-economic backgrounds.
- ► Weight was objectively assessed and we trained over 100 midwives from a large area of central England to test the intervention in routine practice.
- ► We achieved 77% follow-up for the primary outcome but only around 42% of women completed the end of pregnancy follow-up questionnaires.
- ► Although we assessed intervention fidelity, our data on the intervention group were incomplete, with only 65% of weight charts available.

gestational weight gain. There was no evidence of psychological harm.

**Trial registration number** ISRCTN67427351

## INTRODUCTION

In developed countries around 40%–60% of women gain more weight while pregnant than the US Institute of Medicine (IOM) guidelines advise.[1–3] Excessive gestational weight gain is associated with adverse pregnancy outcomes and later obesity.[4 5] Although excessive gestational weight gain is common, no country has an evidence-based intervention to prevent it which can be used in routine care, and there is no global consensus about whether weighing during pregnancy prevents excessive gestational weight gain.[6] Most randomised controlled trials (RCTs) to date have focused on specialist interventions

for obese women who are pregnant, but most women who become pregnant are healthy or overweight but not obese. Pregnancy may be the time when weight control slips and preventative interventions are needed for these women to reduce long-term health risks and potential adverse effects on the infant.

Previous trials involving pregnant women have not found regular weighing either by maternity health professionals within antenatal care or by women themselves to be effective in reducing excessive gestational weight gain, although these trials have been small and/or reported intervention contamination or experienced low adherence to the intervention.[7–10] Collectively, it highlights the need for additional high-quality trials to evaluate interventions that are embedded into routine clinical care. While advice on regular weighing during pregnancy and on optimal weight gain is part of standard antenatal care for pregnant women in many developed countries (eg, USA, Canada, France), this is not the case in many other similar countries (eg, Australia, New Zealand and Netherlands).[6] In England, the National Institute for Health and Care Excellence (NICE) do not recommend it because of a lack of evidence of effectiveness and concerns about the potential for psychological harm.[11] Thus we had the opportunity to test the effectiveness of introducing weighing into routine antenatal care in an environment where it is not the norm.

### Trial development and aims
In preparation for the current study we conducted a feasibility RCT (POPS) to test the acceptability of an intervention where community midwives weighed women and set maximal gestational weight gain limits.[12] The recruitment rate was high at 94% demonstrating that women were very keen to participate in the study. The feasibility trial also included two embedded qualitative studies (interviews) with both women and community midwives. Most women felt that the intervention was useful in encouraging them to think about their weight and believed that it should be part of routine antenatal care. The community midwives commented that the intervention could be implemented within routine care without adding substantially to consultation length. Following our feasibility study, our aim in this trial (POPS2) was to investigate the effectiveness of a behavioural brief intervention based on target setting, routine antenatal weighing and feedback in preventing excessive weight gain.[12 13] It was compared with usual maternity care.

## MATERIALS AND METHODS
### Trial design and population
POPS2 was randomised clinical trial (RCT) with individual randomisation. The trial protocol has been published previously.[13]

### Participant identification and recruitment
#### Participants
Pregnant women under the care of four maternity centres in England were recruited. Women received written information about the study and, if eligible, they were approached after their routine dating scan at 10–14 weeks gestation. Women were eligible if they were confirmed as having a singleton pregnancy with a body mass index (BMI) $\geq 18.5$ kg/m$^2$ at recruitment, expected to receive community midwife led care or shared care (midwife and consultant led care) at recruitment, were aged $\geq 18$ years and between $10^{+0}$ to $14^{+6}$ weeks gestation at recruitment. Women were not eligible if they were unable to understand English or provide informed consent, they attended a weight management programme, they experienced severe mental illness or they were dependent on illicit drugs or alcohol.

### Randomisation and masking
The randomisation list was created by an independent statistician using nQuery Advisor V.7.0. Randomisation was stratified by BMI category at recruitment (healthy weight/overweight/obese) and recruitment site. Participants were individually randomised using random permuted blocks of mixed size (2, 4 or 6). Due to the nature of the intervention, it was not possible to blind participants or community midwives to the intervention. The trial statistician remained blinded to group allocation until completion of analyses. Participants were allocated to the groups by a clinical trials unit telephone randomisation service. Allocation was revealed to researchers by calling the randomisation line.

### Primary outcome
The primary outcome was the proportion of women who exceeded the upper limit of the IOM guidelines for healthy weight gain at 38 weeks gestation defined by their BMI-appropriate weight chart. Weight gain at 38 weeks was calculated as weight at 38 weeks minus pre-pregnancy weight. As many women did not know their pre-pregnancy weight, we assumed that they had gained weight along the median of the healthy weight gain zone defined on the IOM chart. Thus assumed weight gain depended upon BMI category and gestation. The primary endpoint, 38 weeks of pregnancy, was defined as weight recorded after 37 weeks of pregnancy. Births before 37 weeks were classed as pre-term and excluded from weight related analyses.

### Secondary outcomes
Secondary weight-related outcomes were the proportion of women who were within the IOM guidelines for their early pregnancy BMI category at 38 weeks of pregnancy defined by their chart; proportion of women who were below the IOM guidance for healthy weight, weight gain (kg) per week of pregnancy from baseline to end of pregnancy, defined as change in weight (kg) by gestational weeks and weight gain (kg) from baseline to 38 weeks gestation. Other secondary outcomes were change in depression and anxiety between baseline and 38 weeks measured by the Hospital Anxiety and Depression Scale (HADS), physical activity measured by the

Physical Activity in Pregnancy Questionnaire and diet quality measured by the Southampton Food Frequency Questionnaire.[14–16] We also recorded pregnancy-related health outcomes, principally to contribute to future meta-analyses.

### Baseline assessment and follow-up of outcomes

All participants were weighed only in light clothing at baseline by the research team using calibrated scales and had their height measured. In England, midwives see women at 38 weeks gestation and all participants were weighed then for the primary outcome. As a measure of intervention contamination, the usual care (UC) group were asked the question "Did your midwife talk to you about your weight at your last two appointments?"

### Intervention

The intervention supplemented usual antenatal care and was based on self-regulation theory.[17] Self-regulation has been described as a process that has three distinct stages: self-monitoring, self-evaluation and self-reinforcement. Self-monitoring is a method of systematic self-observation, periodic measurement and recording of target behaviours with the goal of increasing self-awareness. The awareness fostered during self-monitoring is considered an essential initial step in promoting and sustaining behaviour change.

We aimed for the intervention to make minimal demands on midwives' time as this would be key to future implementation in routine care. The IOM guideline was the only one available for healthy pregnancy weight gain at the time of the study and we used it to set the limits for weight gain for the intervention group.[1] In UK antenatal care, there are typically eight antenatal consultations and we scheduled for the intervention to take place in each one. Midwives weighed women at each antenatal appointment using calibrated portable weighing scales. Midwives plotted women's weight on an IOM weight chart appropriate to a participant's BMI category at recruitment (figure 1). The chart was attached to the woman's handheld pregnancy notes and outlined a maximum weight gain limit for the next appointment. The published protocol explains how these maximum limits were set.[13] The goal was for weight gain to follow the midpoint line on the chart (figure 1).

At subsequent appointments, midwives gave women feedback on weight gain in relation to the limit, set a new limit for the next appointment and reinforced the value and importance of healthy weight gain. Midwives never asked women to lose weight. Instead, midwives set targets to bring women back towards the centre line (figure 1). Midwives encouraged women to weigh themselves weekly and to record it on the chart, to calculate their weekly weight gain limits and to check progress against the chart. Midwives offered brief advice about healthy eating and exercise in pregnancy.[18]

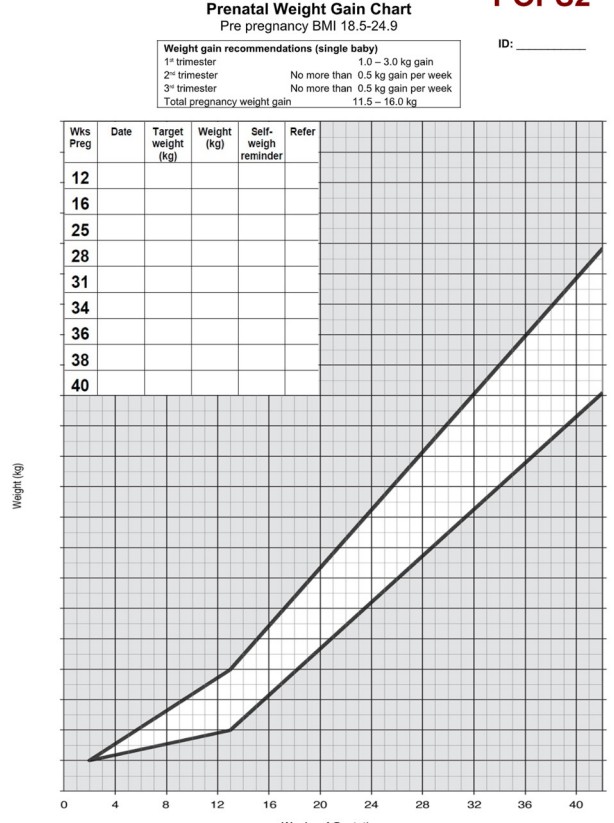

**Figure 1** Weight chart.

### Training of community midwives

The research team trained midwives to deliver all components of the intervention as detailed in the published study protocol.[13] Mindful that only interventions requiring a short training course would ever be widely implemented in routine antenatal care, we designed a 60–70 min module delivered in a group setting to community midwives. A training manual was also developed which included information on study eligibility criteria, recruitment procedures, the importance of adhering to protocol and not contaminating the UC group. Information on the effects of weight gain during pregnancy, instructions about how to weigh and plot weight on the IOM weight chart and how to give feedback on the weight gain chart and example messages were also outlined. Explanation of how to set weight gain limits using the charts and examples of educational and motivational messages that should be given about gestational weight gain, diet and physical activity during pregnancy were also included. Midwives also practiced completing the weight gain charts using prepared case studies.

### UC group

The UC group received standard maternity care and no other intervention. As the intervention did not involve giving lifestyle advice, we did not ask community midwives to refrain from offering usual advice about diet and exercise early in pregnancy to the UC group.

## Patient and public involvement

Feedback from pregnant women and community midwives, which was obtained by a previous feasibility study, was integrated into the design and conduct of this study. In addition, feedback from a maternity patient and public involvement group at local hospital (PRIME) was also incorporated into the design of this study. Participants in this study were not involved in the study recruitment processes. Study participants received a summary of the study results once it was completed.

## Sample size

Six hundred and ten women (305 per group) were sufficient to detect a 15% points difference between the groups (45% vs 60%) in the proportion of women who exceeded the IOM guideline for gestational weight gain with 90% power and 5% significance.[1–3] The sample size included allowance for 20% loss to follow-up. This sample size would also be sufficient to detect 1.6 kg group difference in mean weight gain at follow-up (SD of weight change of 5.5 kg from our feasibility RCT,[12] 90% power, 5% significance level). The sample size was not inflated for clustering by midwife because clustering of this nature does not inflate the type 1 error rate as described elsewhere.[19]

## Statistical analysis

All analyses were performed using an intention-to-treat (ITT) approach, whereby participants were analysed according to randomisation group, with the following predefined exclusions: women who experienced pregnancy loss excluded from all analyses and women who experienced a preterm birth excluded from analyses of weight outcomes. The primary analysis, comparing the proportion exceeding the IOM guidelines between the groups, was undertaken using generalised linear mixed modelling with imputation and the intervention effect presented as OR with corresponding 95% CI. We anticipated that the missingness mechanism for the majority of those with missing weights in the UC group would be related to births taking place outside the due date and unrelated to their weight, that is, missing at random, therefore missing follow-up weights were imputed using multiple imputation via PROC MI in SAS using five replications with group allocation, site, age, ethnicity, Index of Mulitple Deprivation (IMD) quartile, BMI, baseline weight and final weight and gestation, as predictors. Weights was considered missing if a self-reported weight was not available, or they were not considered missing if measured before 37 weeks and if delivery was not preterm. The primary analysis was adjusted for BMI category and site as fixed effects and midwife as a random effect. A subgroup analysis assessing whether there were differences in treatment effect by BMI category was carried out by including a multiplicative interaction term in the modelling. We reasoned that both midwives and women might be motivated much to avoid excessive weight gain if women were already overweight.

The robustness of the results was examined with a sensitivity analysis of the primary outcome and included: complete case analysis; missing 38 week weights imputed with BMI category-specific mean weight; missing 38 week weights imputed with average weight within BMI category-related IOM threshold. Secondary weight-related outcomes were compared using mixed modelling with multiple imputation and adjustments for BMI category, site and midwife as previously described. Linear mixed modelling was used to compare psychological health and physical activity at the end of pregnancy adjusting for baseline values of the outcome in addition to BMI category, site and midwife.

We also conducted per protocol analyses of the primary outcome. Adherence to the protocol was defined in two ways. First, we considered that women had followed the protocol if they recorded their weight every week on at least 70% of occasions prior to delivery or 38 weeks gestation. Second, we assumed adherence if women had recorded their weights at least five times per week, an approach taken by a similar study.[7] We considered that midwives had followed the protocol if they set a correct weight gain target and subsequent weekly targets for women on 70% of a woman's appointments.

## RESULTS

### Trial flow and characteristics of the population

We approached 1271 women and 816 of them were eligible (64.2%). Of them, 656 women (80.4%) agreed to participate and were randomised, 329 agreed to receive the intervention and 327 agreed to UC (figure 2). Baseline characteristics were similar between the trial groups (table 1). A total of 107 midwives were trained to deliver the intervention. The first participant was randomised in November 2014 and follow-up was completed in December 2015.

Outcomes

There was no evidence of a difference in the proportion of women in the intervention and UC groups who gained excessive weight during pregnancy (intervention 27.6% versus usual care 28.9%, OR: 0.84, 95% CI: 0.53 to 1.33, p=0.46). Complete case analysis and different methods of imputation did not alter the results (table 2). Sub-group analysis showed no evidence that the intervention effect differed by baseline BMI status (p=0.41) (table 2).

There was no evidence of differences in the proportion of women in the groups who gained weight within the IOM guidelines (OR 0.92 95% CI: 0.63-1.32) or less than the minimum IOM guidance (OR 1.26 95% CI: 0.86-1.83) for gestational weight (table 3).

On average women in the intervention group gained 10.3 kg and usual care gained 10.7 kg between baseline and 38 weeks of pregnancy. There was no evidence of a difference in the change in weight (kg) during pregnancy (adjusted mean difference -0.42 kg 95% CI: -1.49-0.64) or the amount of weight gained per week of pregnancy

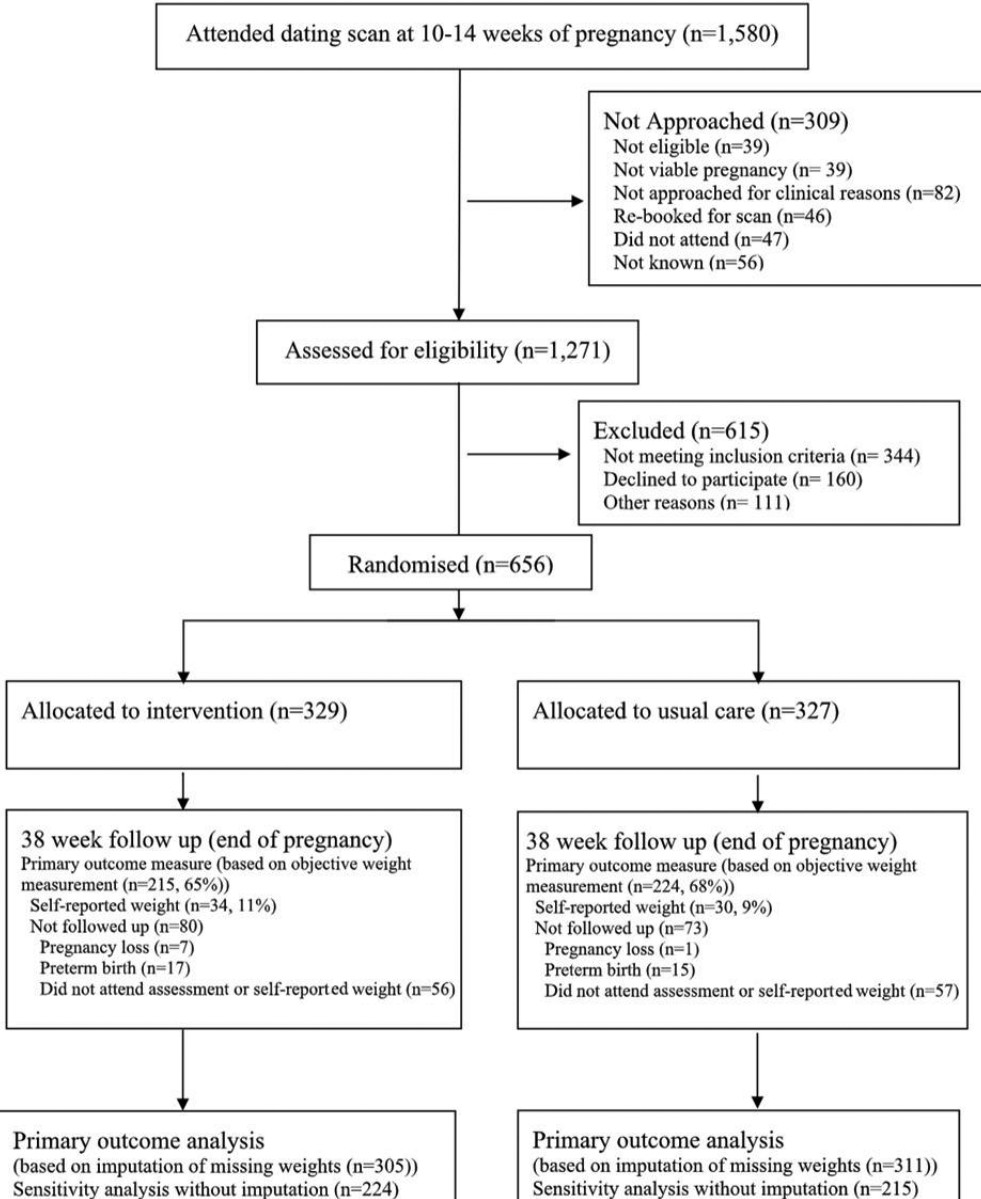

**Figure 2** Trial flow of participants.

between groups (adjusted mean difference -0.01 kg/week 95% CI: -0.038-0.018) (table 4).

Women were doing less physical activity than is recommended for health in pregnancy, and by late pregnancy physical activity had declined, with no difference between groups: mean difference: -4.30 MET hrs/per/week 95% CI:-26.9-18.3 (table 4).

There was no significant difference between groups in anxiety (mean difference -0.58 95% CI:-1.25-0.08) or depression scores (mean difference -0.60, 95% CI:-1.24-0.05) (table 4). We planned to assess dietary quality however issues with both data collection and the scoring algorithm meant we were unable to calculate meaningful summary statistics.

table 4

No serious adverse events were reported. The numbers of pregnancy complications and adverse neonatal outcomes seemed similar in each group (online supplementary table 1). There was no evidence of intervention contamination in the UC group. At follow-up, 17 participants in the UC group responded 'yes' when they were asked if their midwife talked to them about their weight at their last two appointments. The main reasons were because of concern about weight loss, fluid retention, healthy eating advice, large weight gain and reassurance about weight gain.

We obtained 214 (65%) of the weight charts from participants' medical notes. Midwives plotted gestational weights and set weight targets in 57% and 50%, respectively, of scheduled antenatal appointments for the intervention group. Midwives recorded reminding women to weigh themselves weekly at 22% of scheduled appointments. Women in the intervention group weighed themselves on 34% of all weeks. A total of 50.9% (109/214)

## Table 1 Baseline characteristics by group

| Characteristics | Randomisation group | |
|---|---|---|
| | Intervention (N=329) | Usual care (N=327) |
| **Age (years)** | | |
| Age (years) mean (SD) | 29.4 (5.0) | 29.7 (5.2) |
| Range | 18.3–40.6 | 18.0–43.0 |
| **Height (cm)** | | |
| Height (cm) mean (SD) | 163.5 (6.5) | 163.3 (6.7) |
| **Weight (kg)** | | |
| Weight (kg) mean (SD) | 69.3 (13.8) | 69.7 (13.5) |
| **BMI category** | | |
| Healthy | 161 (48.9) | 161 (49.2) |
| Overweight | 106 (32.2) | 103 (31.5) |
| Obese | 62 (18.8) | 63 (19.3) |
| **BMI (kg/m²)** | | |
| BMI (kg/m²) mean (SD) | 25.9 (4.6) | 26.1 (4.8) |
| **IMD quartile** | | |
| IMD quartile 1 (least deprived) | 34/323 (10.5) | 31/325 (9.5) |
| 2 | 60/323 (18.6) | 52/325 (16.0) |
| 3 | 86/323 (26.6) | 96/325 (29.5) |
| 4 (most deprived) | 143/323 (44.3) | 146/325 (44.9) |
| **Ethnicity** | | |
| White | 241/328 (73.5) | 238/327 (72.8) |
| Black Caribbean | 5/328 (1.5) | 6/327 (1.8) |
| Black African | 4/328 (1.2) | 5/327 (1.5) |
| Black Other | 0/328 (0.0) | 1/327 (0.3) |
| Mixed | 12/328 (3.7) | 7/327 (2.1) |
| Chinese | 1/328 (0.3) | 3/327 (0.9) |
| Indian | 10/328 (3.1) | 8/327 (2.5) |
| Pakistani | 34/328 (10.4) | 39/327 (11.9) |
| Bangladeshi | 4/328 (1.2) | 4/327 (1.2) |
| Other Asian | 4/328 (1.2) | 2/327 (0.6) |
| Other | 13/328 (4.0) | 14/327 (4.3) |
| **Marital status** | | |
| Married | 166/315 (52.7) | 195/318 (61.3) |
| Single (living alone) | 48/315 (15.2) | 33/318 (10.4) |
| Single (living with spouse) | 95/315 (30.8) | 86/318 (27.0) |
| Widowed | 0/315 (0.0) | 1/318 (0.3) |
| Divorced/separated (living alone) | 2/315 (0.6) | 2/318 (0.6) |
| Divorced/separated (living with spouse) | 2/315 (0.6) | 1/318 (0.3) |
| **Employment status:** | | |
| In paid employment | 218/319 (68.3) | 236/317 (74.5) |
| Student | 15/319 (4.7) | 5/317 (1.6) |
| Self-employed/freelance | 21/319 (6.6) | 7/317 (2.2) |
| Looking after home/family | 28/319 (8.8) | 24/317 (7.6) |
| Unemployed | 35/319 (11.0) | 40/317 (12.6) |
| Sick/disabled | 1/319 (0.3) | 1/317 (0.3) |
| Retired from paid work | 0/319 (0.0) | 0/317 (0.0) |
| Other | 1/319 (0.3) | 4/317 (1.3) |

## Table 1 Continued

| Characteristics | Randomisation group | |
|---|---|---|
| | Intervention (N=329) | Usual care (N=327) |
| **Smoking status:** | | |
| Current smoker | 27/316 (8.5) | 20/317 (6.3) |
| Number of children mean (SD) n | 0.6 (0.9) 313 | 0.8 (1.1) 315 |
| Attending weight loss programme | 4/317 (1.3) | 5/317 (1.6) |

Figures are n (%) unless otherwise stated.
BMI, body mass index; SD, standard deviation.

of women in the intervention group weighed themselves five times or more, 15.9% (34/214) 2–4 times and 33.2% (71/214) once or less. In the per protocol analyses, there was no evidence of a difference between the groups in the proportions who gained excessive gestational weight (online supplementary table 2).

## DISCUSSION

There was no evidence that the intervention of weight gain limit setting, regular weighing and feedback delivered by community midwives as part of routine antenatal care was effective. There was however no evidence of psychological harm from the intervention. These findings contribute to the current and ongoing debate about whether routine weighing should be re-introduced throughout pregnancy.

Pregnant women have reported that they expect to be weighed during pregnancy and feel that it should be part of routine antenatal care.[12 20] However, three previous trials have investigated the effectiveness of behavioural interventions based on regular weighing to prevent excessive gestational weight gain and none of them were effective.[7–10] In one trial, women were advised by a medical student to weigh themselves seven times during pregnancy and were given an IOM weight chart and a table for them to assess their own progress against the targets.[9] In the second trial, women spent half an hour discussing the importance of healthy weight gain with a research midwife and were encouraged to weigh themselves serially and given an overall weight target, which they were encouraged to discuss with a clinician.[8] However, antenatal clinicians were not trained to intervene. In the third trial, women were measured in antenatal clinics and their weights were recorded; posters in the clinic were placed for women's motivation to stay within the IOM guidelines.[7] The intervention in the present trial was the most complete behavioural intervention to date, comprising both midwife training for routine weighing, setting weight gain limits and feedback, as well as advice to women to weigh themselves weekly.

The lack of effectiveness may be attributable to poor intervention delivery. Unlike previous trials, we recorded detailed information about intervention fidelity. In our

**Table 2** Comparison of primary outcome (proportion exceeding BMI-related IOM guideline for weight gain during pregnancy) and subgroup analysis

| | Intervention | Usual care | | Intervention–usual care | | |
| | n/N (%) | n/N (%) | Adjusted % difference (95% CI) | Adjusted OR (95% CI) | P value |
|---|---|---|---|---|---|
| **Primary analysis*** Proportion exceeding IOM guideline (multiple imputation of missing 38 week weights) | 81/305 (27.6) | 90/311 (28.9) | −3.5 (−17.8 to 10.7) | 0.84 (0.53 to 1.33) | 0.46 |
| **Sensitivity analysis of proportion exceeding IOM guideline** | | | | | |
| Complete case analysis† | 51/215 (23.7) | 59/224 (26.3) | −4.8 (−19.8 to 10.3) | 0.78 (0.48 to 1.26) | 0.31 |
| Imputation with BMI category-specific mean weight‡ | 85/305 (27.9) | 91/311 (29.3) | −3.1 (−16.0 to 9.8) | 0.86 (0.59 to 1.26) | 0.44 |
| Imputation with average weight within BMI category-related IOM threshold‡ | 87/305 (28.5) | 93/311 (29.9) | −3.1 (−16.0 to 9.9) | 0.86 (0.59 to 1.25) | 0.44 |

| Subgroup | Intervention | Usual care | | | |
| **BMI category at recruitment** | **Number exceeding IOM guideline/N (%)§** | **Number exceeding IOM guideline/N (%)§** | **Adjusted OR (95% CI)** | **P value (interaction)** | **–** |
|---|---|---|---|---|---|
| Healthy weight | 15/148 (10.3) | 22/161 (13.5) | 0.69 (0.22 to 2.21) | 0.41 | – |
| Overweight | 38/95 (39.8) | 34/93 (36.6) | 1.11 (0.60 to 2.04) | | |
| Obese | 31/62 (50.3) | 34/57 (59.6) | 0.69 (0.30 to 1.58) | | |

Analysis adjusted by Site, BMI category and midwife (random effect).
*Includes objective and self-reported weights; missing weights imputed using multiple imputation; pooled estimates.
†Includes objective weights only.
‡Includes objective and self-reported weight.
§Includes objective and self-reported weights; missing weights imputed using multiple imputation; pooled estimates.
BMI, body mass index; IOM, Institute of Medicine.

feasibility trial, most midwives commented that they felt the intervention was feasible taking on average of about 1–2 min per appointment and it was not perceived as adding substantially to their workload. Midwives also commented that they liked the intervention because it was simple to do and provided them with a legitimate opportunity to raise the topic of gestational weight gain. However, the process evaluation showed only moderate fidelity by midwives in weighing women and setting a target, and little encouragement to women to weigh themselves at home. Only a small proportion of women weighed themselves every week through pregnancy.

Beyond pregnancy, among adults seeking to lose weight, adding regular self-weighing to behavioural weight loss programmes increases effectiveness.[21] The evidence from trials is supported by strong evidence that self-weighing is

**Table 3** Comparison of secondary outcome: proportion within or below BMI-related IOM guideline for weight gain during pregnancy

| | Intervention | Usual care | Intervention–usual care | | |
| | n/N (%) | n/N (%) | Adjusted % difference (95% CI) | Adjusted OR (95% CI) | P value |
|---|---|---|---|---|---|
| Within IOM guideline* | 96/305 (31.5) | 108/311 (34.6) | −2.0 (−14.6 to 10.6) | 0.92 (0.63 to 1.32) | 0.63 |
| Below IOM guideline* | 125/305 (40.9) | 114/311 (36.5) | 4.9 (−7.4 to 17.2) | 1.26 (0.86 to 1.82) | 0.24 |

Analysis adjusted by Site, BMI category and midwife (random effect).
*Includes objective and self-reported weights; missing weights imputed using multiple imputation; pooled estimates.
BMI, body mass index; IOM, Institute of Medicine.

**Table 4** Comparison of secondary outcomes

| | Intervention | | | Usual care | | | Intervention–usual care | |
| --- | --- | --- | --- | --- | --- | --- | --- | --- |
| | Baseline | 38 weeks | Change | Baseline | 38 weeks | Change | Adjusted mean 38 weeks (95% CI)* | P value |
| | Mean (SD) n | Mean (SD) n | Mean (SD) n | Mean (SD) n | Mean (SD) n | Mean (SD) n | | |
| Weight (kg)† | 71.21 (13.62) 305 | 81.49 (14.35) 305 | 10.28 (5.87) 305 | 71.06 (13.19) 311 | 81.79 (14.35) 311 | 10.73 (6.88) 311 | −0.42 (−1.49 to 0.64) | 0.43 |
| HADS: Anxiety | 4.88 (3.50) 313 | 5.18 (3.09) 136 | 0.45 (2.82) 136 | 5.15 (3.28) 318 | 5.89 (3.58) 133 | 0.82 (3.33) 132 | −0.58 (−1.25 to 0.08) | 0.08 |
| HADS: Depression | 3.29 (2.90) 313 | 3.93 (3.04) 136 | 0.75 (2.83) 136 | 3.49 (3.34) 318 | 4.56 (3.04) 133 | 1.29 (3.20) 132 | −0.60 (−1.24 to 0.05) | 0.07 |
| Total physical activity (MET/hours/week) | 283.68 (144.52) 313 | 246.63 (104.97) 136 | −35.02 (115.94) 136 | 278.91 (158.50) 317 | 240.65 (115.26) 132 | −23.43 (117.05) 131 | −4.30 (−26.94 to 18.34) | 0.71 |

*Adjusted by baseline value, site, BMI category and midwife (random effect).
†Includes objective and self-reported weights; missing weights imputed using multiple imputation; pooled estimates.
HADS, Hospital Anxiety and Depression Scale; SD, standard deviation.

a key component of the behavioural repertoire of people who are successful in maintaining their weight.[22] However, a programme based on self-weighing alone was only minimally effective.[21] We had expected that the greater engagement of women in their own health during pregnancy and concern for the health of their baby might make it a moment when regular weighing would prompt other self-regulatory controls and stimulate effective weight management. In the UK, weighing of women routinely during antenatal care is not recommended and this practice is not part of antenatal care in many other countries, though it is routine in others.[11 23] NICE noted the lack of evidence of benefit, but also expressed concerns that weighing may cause psychological harm. There was no evidence to indicate any health harms in this trial and other studies suggest that, far from increasing anxiety, it is welcomed by women.[12 20]

Previous research suggested that in many developed countries, the majority of women gain excess gestational weight; only 28% of the UC group did so here, but not the 60% we assumed would in the sample size calculation. We can only speculate on why the proportion of women gaining excess weight was lower than that was expected in this trial. First, it may be due to contamination, with midwives intervening in some unspecified way among women in the UC group, but we found no evidence on it in feedback from the UC group and there were no weight charts in the notes of UC women. Second, it may be due to the trial enrolled women who were particularly weight conscious, as 25% of eligible women declined to participate. However, all of those who declined would need to have gained excess weight to reach the frequency of weight gain cited in other studies. One of the attractions of this kind of programme is that it is scalable and well-suited to routine care, so that, if effective, it could be applied routinely in prevention in the way similar to few other interventions. Future research will need to identify how to engage midwives and women more actively in the process of self-weighing, consider additional behavioural components or identify other interventions and test their effectiveness in this context.

This study has several strengths. Most (~80%) eligible women participated in the trial, meaning the results reflect the impact in the general population. A relatively large proportion of women were recruited from non-white ethnic groups (27%) and/or low socioeconomic backgrounds (55%) who are often under-represented in trials. Weight was objectively assessed. Rather than recruiting a small number of highly motivated and highly trained midwives, we trained over 100 midwives from a large area of central England to test the intervention in routine practice. To our knowledge, this is the first trial where community midwives have delivered an intervention involving setting weight gain limits, regular weighing, encouraging weekly self-weighing and providing feedback. Unlike trials testing similar interventions, we collected detailed process data on the fidelity of delivery of the intervention and women's adoption of the advice to weigh themselves.

Our findings should also be interpreted in light of some limitations. We estimated that we would follow-up 80% of participants for the primary outcome when calculating the sample size but achieved only 77%. However, only around 42% of women completed the end of pregnancy follow-up questionnaires. Although we assessed fidelity, our data on the intervention group were incomplete, with the availability of only 65% of weight charts. This was because some women experienced miscarriage, their notes were not available to the research team, they withdrew from the trial or they removed the charts from their notes. The proportion of women who gained excessive weight was markedly lower than predicted, 30% actual versus 60% predicted from the literature. The sample size was predicated on having 90% power to detect a 15% absolute risk reduction, a relative reduction in incidence of 25%. However, a 25% relative reduction from 30% would imply a smaller absolute difference, thus reducing the power of the study below that originally envisaged, which means that a benefit of this treatment programme cannot be confidently ruled out. The development of our intervention may have been enhanced with co-creation with midwives, although the intervention was refined based on the feedback from midwives in our feasibility study.

## CONCLUSION

We did not find evidence to support the value of setting a maximum weight gain limit, regular weighing and feedback during pregnancy to prevent excessive gestational weight gain. The trial provides reassurance that weighing is not harmful, but in countries where regular weighing is part of usual maternity care women should be advised that other strategies may be required to prevent excessive gestational weight gain.

**Author affiliations**
[1]School of Sport, Exercise and Health Sciences, Loughborough University, Loughborough, UK
[2]Institute of Applied Health Research,College of Medical and Dental Sciences, University of Birmingham, Birmingham, UK
[3]Nuffield Department of Primary Care Health Sciences, University of Oxford, Radcliffe Observatory Quarter, Oxford, UK
[4]Nuffield Department of Women's and Reproductive Health, University of Oxford, John Radcliffe Hospital, Oxford, UK
[5]Population Health Sciences, Bristol Medical School, University of Bristol, Canynge Hall, Bristol, UK
[6]Sherwood Forest Hospitals NHS FoundationTrust, Nottinghamshire, UK

**Acknowledgements** We would like to thank the women who took part in this research. We would also like to thank the hospitals and community midwives who agreed to participate in this research; these women were from Birmingham Women's NHS Foundation Trust; The Dudley Group NHS Foundation Trust; Oxford University Hospitals NHS Trust and South Warwickshire NHS Foundation Trust. We would also like to thank the PRIME PPI group at Birmingham Women's Hospital for their comments and feedback on this study.

**Contributors** AD conceived the original idea for the study with input from PA, KJ, SAJ, AL, LM, SC, CM and SK. AD wrote the protocol with contribution from the other authors. SC was responsible for overseeing data collection. CO assisted with data collection and provided clinical advice. AR was responsible for overseeing the writing of the statistical analysis plan. MU conducted the analyses. All authors

had full access to the data, taken responsibility for the integrity of the data and the accuracy of the data analysis, contributed to the interpretation of the results and reviewed and approved the final manuscript. AD drafted the article and all other authors commented on this draft. AD is the guarantor.

**Funding** The National Institute of Health Research School for Primary Care Research part funded this study. NIHR CLAHRC-Oxford and NIHR CLAHRC-West Midlands also part funded this study. KJ, CM and SK are part funded by the National Institute for Health Research (NIHR) Collaboration for Leadership in Applied Health Research and Care (CLAHRC) West Midlands. SAJ and PA are NIHR Senior Investigators and supported by the Oxford NIHR Biomedical Research Centre and NIHR CLAHRC Oxford. AR is supported by NIHR Oxford Biomedical Research Centre. LM works part time for Sensyne Health plc.

**Disclaimer** The views expressed in this publication are not necessarily those of NHS, the NIHR or the Department of Health. The study was externally peer-reviewed for scientific quality. The funders had no involvement in conducting the research.

**Competing interests** None declared.

**Patient consent for publication** Not required.

**Ethics approval** Ethical approval was obtained from NRES Committee West Midlands—South Birmingham: 14/WM/1134, 02/10/14. All procedures performed in this study were in accordance with the ethical standards of the institutional review board, the American Psychological Association and the 1964 Helsinki Declaration. Written informed consent was obtained for all individual participants included in this study.

**Provenance and peer review** Not commissioned; externally peer reviewed.

**Data availability statement** Data are available upon reasonable request.

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
