## [Reviewer comments · BMJ Open]

ARTICLE DETAILS

TITLE (PROVISIONAL)	Effectiveness of a behavioural intervention involving regular weighing and feedback by community midwives within routine antenatal care to prevent excessive gestational weight gain: POPS2 randomised controlled trial
AUTHORS	Daley, Amanda; Jolly, Kate; Jebb, Susan; Roalfe, Andrea; Mackillop, Lucy; Lewis, Amanda; Clifford, Sue; Usman, Muhammad; Ohadike, Corah; Kenyon, Sara; MacArthur, Christine; Aveyard, Paul

VERSION 1 - REVIEW

REVIEWER	Helen Skouteris Monash University, Australia
REVIEW RETURNED	21-Mar-2019

GENERAL COMMENTS	The aim of this study was to assess the effectiveness of a brief behavioural intervention based on routine antenatal weighing to prevent excessive gestational weight gain (GWG). Preventing excessive GWG is an important public health issue. Involving community midwives in an intervention to prevent excessive GWG is a practical solution that needs to be rigorously explored. I have several reservations about the study however and they are: 1. The intervention does not appear to have been co-designed with midwives. Midwives in a feasibility study said the intervention was useful and could be implemented into routine care. Please add the lack of co-design as a limitation to the discussion and the implications of this limitation moving forward in relation to how the intervention now needs to be tailored to lead to better adherence by midwives and women.2. How many midwives were trained; where did the training take place; did all the midwives in the study take part in the training; how were the learning outcomes of the training assessed; that is, how did you know whether the midwives had completed the training successfully; why was the training not co-designed with midwives?3. Did you collect data on what advice the midwives gave to the women; do you know whether they actually gave advice; how was fidelity to the whole intervention delivery measured as opposed to just completion of the weight charts; how were health behaviour change strategies delivered by the midwives measured to ensure they were not just providing healthy eating and exercise
---

	information alone (we know information alone does not work) - this point is particularly important because you have included PA as a secondary outcome and yet it is not clear how the midwives motivated/encouraged etc women to increase their PA during pregnancy for weight gain management? 4. I do not follow the inclusion of the mental health secondary outcomes; please justify their inclusion based on theory in the introduction. Also, why did you not include healthy eating as a secondary measure? How reliable was your measure of PA? What were the Cronbach's alphas for the secondary measures you used?
--	---

REVIEWER	Mary-Ann Davey Monash University, Australia
REVIEW RETURNED	01-Apr-2019

GENERAL COMMENTS	This well-conducted RCT addresses an important problem in maternity care that affects a large number of women and adversely affects perinatal outcomes. Eligibility criteria were appropriately inclusive, and the high participation rate by eligible women, representative of the diverse population, indicates that women saw the intervention as acceptable. It is however disappointing that the primary end-point was available for only 77% of participants. The low response to the survey is not unusual but means we cannot be confident in the generalisability of its results. A good process evaluation was planned and found sub-optimal compliance on the part of women and midwives despite pilot studies indicating that both groups supported the intervention. The detected proportion of women in the control group who gained excessive weight was half that anticipated and the reason for this discrepancy remains unknown. I have a few questions or areas that need clarification: Methods The randomisation process is well-designed and carefully described The details of the primary outcome need to be more fully described. Please clarify what P6 lines 37-42 means. It indicates that we need more precise information on the calculation of the primary outcome. Is it (weight at 38 weeks- reported pre-pregnancy weight)? Or weight at 38 weeks-measured weight at recruitment? If they gave birth before 38 weeks, how was this adjusted? Also, P6, line 51 - does 'proportion who were below healthy weight' mean 'gained less than the lower limit of IOM recommendation? Please clarify this and the rest of that sentence. I would find it difficult to replicate as currently described. It would help to describe how depression, anxiety and physical activity were measured. P8. The 'patient involvement' paragraph is confusing. Does it mean 'Consumer involvement'? I'm not sure what is meant by the last 2 sentences, perhaps because it is not clear who 'participants' are here – are they participants in the pilot studies and consumer consultation processes? Does the 'summary of the study' refer to the protocol or the results?
--

	Sample size. My calculations indicate that a sample size of 610 does not include the inflation for the intracluster correlation – it would have been 854. It appears that a similar oversight was made with regard to the power for the secondary outcome described. Though after the event it is clear that a larger sample would not have affected the primary outcome; however it could have affected the depression and anxiety results. Statistical analysis It would help to know how many were excluded because of preterm birth? Please confirm (and amend) that the variable ‘final weight’ was included along with the other specified variables in the multiple imputation command. Why were so many final weights missing? Please clarify why you needed to adjust for BMI category in the primary outcome since the groups look similar at baseline. Did you consider adjusting for marital status given the difference between groups? Line 31 – are the co-variables the same as those used for multiple imputation. Results P10. Line 56 - 816 is not 54.8% of 1271, and 656 is not 75.8% of 816. Please clarify or correct. Please present the results related to contamination in text or table. Typo P5, Line 56 should be 14+6 not 14-6
--	---

REVIEWER	David Blanco Universitat Politècnica de Catalunya
REVIEW RETURNED	01-Apr-2019

GENERAL COMMENTS	This report shows the results of an evaluation of the consistency between the CONSORT checklist you submitted and the information that was reported in the manuscript. Please, make the following revisions:  • For CONSORT Item 6a ("Completely defined pre-specified primary and secondary outcome measures, including how and when they were assessed"), please specify how and where some secondary outcomes (depression, anxiety, physical activity, and pregnancy-related health) were evaluated. Presenting the study outcomes transparently makes the study results more straightforward to understand. • For CONSORT Item 11a ("If done, who was blinded after assignment to interventions (for example, participants, care providers, those assessing outcomes) and how"), I suppose that due to the nature of the intervention it was not possible to blind participants. Please, include a sentence in the end of the "Randomisation and masking" subsection making that point explicit. Also, please comment on the blinding status of the outcome assessors.
---

VERSION 1 – AUTHOR RESPONSE

REVIEWER 1

The intervention does not appear to have been co-designed with midwives. Midwives in a feasibility study said the intervention was useful and could be implemented into routine care. Please add the lack of co-design as a limitation to the discussion and the implications of this limitation moving forward in relation to how the intervention now needs to be tailored to lead to better adherence by midwives and women.

RESPONSE: We completed a feasibility trial and qualitative interviews with community midwives and we incorporated their experiences of delivering the intervention within usual antenatal care when planning this trial. One of the co-investigators was a midwife and her views were incorporated in to the design of the trial and the intervention. Comments in line with the reviewer suggestions have been added. See page 21.

How many midwives were trained; where did the training take place; did all the midwives in the study take part in the training; how were the learning outcomes of the training assessed; that is, how did you know whether the midwives had completed the training successfully; why was the training not co-designed with midwives?

RESPONSE: 107 midwives were trained as stated on page 11. Midwives had to complete the training in order to take part in the study. Midwives participated in a learning exercise at the end of the training session to ensure they understood the study processes and this was checked by researchers. The training was based on feedback received from midwives in the feasibility study.

Did you collect data on what advice the midwives gave to the women; do you know whether they actually gave advice; how was fidelity to the whole intervention delivery measured as opposed to just completion of the weight charts; how were health behaviour change strategies delivered by the midwives measured to ensure they were not just providing healthy eating and exercise information alone (we know information alone does not work) - this point is particularly important because you have included PA as a secondary outcome and yet it is not clear how the midwives motivated/encouraged etc women to increase their PA during pregnancy for weight gain management?

RESPONSE: We did collect data on what advice midwives gave to women in both groups and this will be reported in a separate publication and in a corresponding qualitative study. However, it is important to highlight that the main components of the intervention were the setting of weight gain limits and regular self weighing by women and we reported the fidelity of these components in the manuscript.

I do not follow the inclusion of the mental health secondary outcomes; please justify their inclusion based on theory in the introduction. Also, why did you not include healthy eating as a secondary measure? How reliable was your measure of PA? What were the Cronbach's alphas for the secondary measures you used?

RESPONSE: NICE do not recommend that pregnant women are weighed during pregnancy because of concerns that doing so may cause psychological harm. With this guidance in mind it was important

and appropriate that we assessed whether levels of psychological health were different in the groups. We did measure diet quality but due to issues with the quality of the study data this was not included and we have added a comment on this at the end of the results section. The reliability of the secondary outcomes is referred to in the references provided.

REVIEWER 2

The details of the primary outcome need to be more fully described. Please clarify what P6 lines 37-42 means. It indicates that we need more precise information on the calculation of the primary outcome. Is it (weight at 38 weeks- reported pre-pregnancy weight)? Or weight at 38 weeks-measured weight at recruitment? If they gave birth before 38 weeks, how was this adjusted?

RESPONSE: The primary endpoint, 38 weeks of pregnancy, was defined as weight recorded after 37 weeks of pregnancy. Births before 37 weeks were classed as preterm. Gestational weight gain was defined as (weight at 38 weeks - pre pregnancy weight). For pre pregnancy weight it was assumed that women had gained the average weight at recruitment in line with their gestation according to BMI category (Institute of Medicine) i.e. (weight at 38 weeks – weight at time of recruitment) + average BMI-specific gestational weight gain at recruitment. We have clarified this in the methods section (see page 6).

Also, P6, line 51 - does 'proportion who were below healthy weight' mean 'gained less than the lower limit of IOM recommendation? Please clarify this and the rest of that sentence. I would find it difficult to replicate as currently described. It would help to describe how depression, anxiety and physical activity were measured.

RESPONSE: The proportion who were below healthy weight meant they gained less than the lower limit of IOM recommendation for gestational weight gain within their BMI category. We have changed the text to make this clearer to the reader (see page 7). Anxiety and depression were measured using HADS. Physical activity was measured using the Physical Activity in Pregnancy Questionnaire; these outcomes are also outlined in our protocol. We have added this information to the methods section of the manuscript (page 7).

P8. The 'patient involvement' paragraph is confusing. Does it mean 'Consumer involvement'? I'm not sure what is meant by the last 2 sentences, perhaps because it is not clear who 'participants' are here – are they participants in the pilot studies and consumer consultation processes? Does the 'summary of the study' refer to the protocol or the results?

RESPONSE: We have tried to make this section clearer. We are reporting PPI involvement in line with the requirements of the journal.

My calculations indicate that a sample size of 610 does not include the inflation for the intracluster correlation – it would have been 854. It appears that a similar oversight was made with regard to the power for the secondary outcome described. Though after the event it is clear that a larger sample would not have affected the primary outcome; however it could have affected the depression and anxiety results.

RESPONSE: Apologies this was an error in the description of the sample size methodology. The final design was not inflated for clustering since the midwife clustering occurred before randomisation and

therefore can be classed as ignorable (Kahan & Morris, 2013). Although not required, we have adjusted for midwife effect in the analysis to increase the power. We have corrected the sample size description and added this reference. Kahan BC, Morris TP. Assessing potential sources of clustering in individually randomised trials. BMC Medical research methodology 2013;13:58. See page 9.

It would help to know how many were excluded because of preterm birth?

RESPONSE; Thirty-two participants were excluded due to preterm birth. This information is included in Figure 1.

Please confirm (and amend) that the variable 'final weight' was included along with the other specified variables in the multiple imputation command.

RESPONSE: Final weight was included in the imputation analysis. We have clarified this in the methods section. See page 10.

Why were so many final weights missing?

RESPONSE: We obtained a 77% follow up, only 3% less than anticipated. This was mostly because of preterm births, missed appointments at 38 weeks gestation and preterm births.

Please clarify why you needed to adjust for BMI category in the primary outcome since the groups look similar at baseline. Did you consider adjusting for marital status given the difference between groups?

RESPONSE. The randomisation was stratified by BMI category as we believed that weight gain was associated with BMI. The analyses were therefore adjusted by BMI in line with the trial design and following recommendations in the statistical literature (Kahan & Morris, 2011) which are that if randomisation is stratified, this variable is adjusted for in the analysis. Marital status was not considered a priori as a variable that would affect maternal weight gain therefore we did not consider it appropriate to adjust for this in our analyses. Instead we followed our pre-specified analysis plan.

Kahan BC & Morris TP. Improper analysis of trials randomised using stratified blocks or minimisation. *Statistics in Medicine* 2011. 31;4:328-340.

Line 31 – are the co-variates the same as those used for multiple imputation.

RESPONSE: The covariates referred to here were BMI, site and midwife. We have amended the text clarify this. See page 11.

P10. Line 56 - 816 is not 54.8% of 1271, and 656 is not 75.8% of 816. Please clarify or correct.

RESPONSE: Apologies, these percentages have now been corrected. See page 11.

Please present the results related to contamination in text or table.

RESPONSE: We have presented the data from question that assessed the possibility of contamination in the usual care group at the end of the results section.

P5, Line 56 should be 14+6 not 14-6

RESPONSE: Thank you – we have corrected this error.

REVIEWER 3

For CONSORT Item 6a ("Completely defined pre-specified primary and secondary outcome measures, including how and when they were assessed"), please specify how and where some secondary outcomes (depression, anxiety, physical activity, and pregnancy-related health) were evaluated. Presenting the study outcomes transparently makes the study results more straightforward to understand.

RESPONSE: We have added further detail about our secondary outcomes (page 7) and these are also listed in the study protocol.

For CONSORT Item 11a ("If done, who was blinded after assignment to interventions (for example, participants, care providers, those assessing outcomes) and how"), I suppose that due to the nature of the intervention it was not possible to blind participants. Please, include a sentence in the end of the "Randomisation and masking" subsection making that point explicit. Also, please comment on the blinding status of the outcome assessors.

RESPONSE: We have added the comments suggested by the reviewer (Page 6). We had already commented on the blinding of the assessor so have not added any further comments.

VERSION 2 – REVIEW

REVIEWER	Mary-Ann Davey Monash University, Australia
REVIEW RETURNED	27-Jul-2019

GENERAL COMMENTS	Thank you for answering all of my questions and modifying the paper accordingly. This paper is commendable in its transparency and understated honesty. Congratulations!
---